# Scalable modular architecture for universal quantum computation

Fernando Gago-Encinas[1, *] and Christiane P. Koch[1, †]

[1]*Freie Universität Berlin, Fachbereich Physik and Dahlem Center for Complex Quantum Systems, Arnimallee 14, 14195 Berlin, Germany*

(Dated: September 26, 2025)

Universal quantum computing requires the ability to perform every unitary operation, i.e., evolution operator controllability. In view of developing resource-efficient quantum processing units (QPUs), it is important to determine how many local controls and qubit-qubit couplings are required for controllability. Unfortunately, assessing the controllability of large qubit arrays is a difficult task, due to the exponential scaling of Hilbert space dimension. Here we show that it is sufficient to connect two qubit arrays that are evolution operator controllable by a single entangling two-qubit gate in order to obtain a composite qubit array that is evolution operator controllable. The proof provides a template to build up modular QPUs from smaller building blocks with reduced numbers of local controls and couplings. We illustrate the approach with two examples, consisting of 10, respectively 127 qubits, inspired by IBM quantum processors.

## I. INTRODUCTION

The development of quantum hardware is making fast progress over the past few years, bringing us from the NISQ era [1] closer to fault tolerance [2] and practical quantum error correction [3, 4]. At the forefront are architectures based on trapped neutral [4, 5] and ionic [2, 6, 7] atoms as well as superconducting qubits [3, 8–12], all of which are expected to scale up further. Important landmarks such as the implementation of fault-tolerant operations [2] or demonstrations of break-even for quantum error correction [3, 4] and quantum simulation [7, 13] shift the focus to larger devices. Typically, scalable designs are obtained via a modular approach, linking together multiple smaller quantum processors with limited connectivity rather than developing larger processors [14–17].

One requirement for realizing quantum computing is the ability to perform every unitary operation [18]. In control theory, this is termed evolution operator controllability [19]. Controllability analysis can thus be used as a valuable tool in quantum chip design to ensure that all operations on the device are possible while at the same time simplifying the device [20, 21]. The same concepts that are at the core of controllability analysis can also be employed to the expressivity of parameterized quantum circuits [21, 22]. Controllability tests can be used to identify the sweet spot balancing universality and the complexity of the system's architecture [20] where complexity refers to qubit-qubit couplings and external qubit controls, i.e., the resources to connect and alter the qubits. Reducing the amount of built-in resources while maintaining the computational capability of the quantum device has not yet been a focus for device design but will be important for further scaling up devices, as is evident just

by looking at the number of classical control lines needed to operate today's most advanced QPUs [2–6, 8–10].

Controllability can be rigorously proven by studying the dynamical Lie algebra of the system, i.e., the algebra generated by a system's Hamiltonian split into its time-independent drift and all the control operators [19]. Full rank of the dynamical Lie algebra implies that the system is evolution operator controllable [19]. However, only in exceptional cases can the dynamical Lie algebra be calculated analytically [23–28]. Alternatively, one can determine the rank of the dynamical Lie algebra numerically. The most straightforward way would be to construct a complete basis of the dynamical Lie algebra (which is also a vector space) but this becomes unstable in the required orthogonalization step already for a relatively small number of qubits. Construction of the full algebra can be circumvented by graph methods [20, 29]. Nevertheless, this approach will eventually fail too, due to the exponential scaling, with the number of qubits, of the size of Hilbert space and thus the dimension of the graph. While the curse of dimensionality can — in principle — be evaded by carrying out the controllability test via a parametric quantum circuit in a hybrid approach combining classical optimization with a quantum device [21], it remains yet to be seen how reliably the dimensional expressivity test at the core of this method can be carried out on noisy quantum devices and for larger numbers of qubits. Thus, at present, controllability tests that are generally applicable as well as demonstratedly reliable and scalable are missing. A change in perspective is made possible when proving controllability of a large quantum system from the controllability of its constituent subsystems [28, 30, 31].

Here, we leverage this change in perspective for quantum chip design. Instead of attempting to prove controllability for all qubits at once, we show how to build up controllable qubit arrays from smaller, fully controllable subsystems. We can start from building blocks as small as two (or a few) qubits where it is straightforward to prove controllability. Surprisingly, a single controllable entangling link between two qubits in each of

———

* Present address: Deutsches Elektronen-Synchrotron DESY, Platanenallee 6, 15738 Zeuthen, Germany
† christiane.koch@fu-berlin.de

the quantum processing subunits is sufficient. Such links are routinely realized in current quantum computing architectures, for example via tunable couplers [32–34] for superconducting qubits. While, for a given number of qubits, our approach will not result in the QPU with the smallest possible number of local controls and qubit-qubit couplings, it allows for designs with significantly reduced resources. As an example, we first discuss how two five-qubit arrays that are connected via a tunable coupler result in a controllable ten-qubit array. We then show that this method can be scaled up to generate large devices, thus opening a new route for more resource-efficient device design.

## II. CONTROLLABILITY OF BIPARTITE SYSTEMS WITH AN ENTANGLING CONTROL

We briefly recap evolution operator controllability of a quantum system [19]. Consider a system with Hilbert space dimension $\dim(\mathcal{H}) = n$ whose Hamiltonian can be written as

$$\hat{H}(t) = \hat{H}_0 + \sum_{j=1}^{m} u_j(t)\hat{H}_j \tag{1}$$

for some time-independent drift $\hat{H}_0$, time-dependent external controls $u_j(t)$ and linearly coupled control operators $\hat{H}_j$. The system is said to be evolution operator controllable if, for any target unitary $\hat{U}_{tgt} \in U(n)$, there exist controls $u_j(t)$, a final time $T$ and a phase $\varphi \in [0, 2\pi]$ such that the time evolution generated by $\hat{H}$ at time $T$ with controls $u_j$ verifies $\hat{U}(T, u_1, ..., u_j) = e^{i\varphi}\hat{U}_{tgt}$. In other words, a system is said to be controllable if any unitary operation can be implemented at some final time up to a global phase.

Now, we show how to ensure that controllability is maintained when assembling a larger system by connecting smaller controllable systems. The main idea is rather simple. Assume that we have two, possibly different, qubit arrays ("modules") with given configurations of qubit-qubit couplings and local qubit controls, that are both evolution operator controllable. Then it suffices to connect the two arrays with a control acting on two qubits, one in each module, that is capable of generating entanglement between them, for the resulting multipartite system (composed of the original arrays and the new entangling control) to also be evolution operator controllable. We can keep connecting modules to the original system with one entangling operation per new module and thus design an arbitrarily large system that is evolution operator controllable.

Note that ignoring local contributions, a two-qubit coupling between the $\mu$-th qubit in module $\mathcal{A}$ and the $n$-th qubit in module $\mathcal{B}$ can be in general written as

$$\hat{H}_c^{\mu,n} = \sum_{\alpha,j=1}^{3} c_{\alpha,j}\hat{\sigma}_\alpha^{(\mu)} \otimes \hat{\sigma}_j^{(n)}, \tag{2}$$

where the coefficients $c_{\alpha,j}$ are real-valued, with at least one of them nonzero. We can then state the main theorem formally as follows:

**Theorem 1.** *Let $\mathcal{A}$ and $\mathcal{B}$ be two evolution operator controllable qubit arrays with $M$ and $N$ qubits. Let $\hat{H}_c^{\mu,n}$ be a two-qubit operator given by Eq. (2). Then, the extended bipartite system with a tunable two-qubit coupling with control operator $\hat{H}_c^{\mu,n}$ is operator controllable.*

This result is a particular case of more general results on controllability of multipartite systems [30, 31]. Our proof of Theorem 1, presented in Section A, builds a full basis of the Lie algebra space. This new constructive method provides further information about the depth of commutators required to implement certain elements in the Lie algebra, which can in principle be related to how easily some unitary operations can be implemented on the global system.

Considering the simplest example from a gate-oriented perspective, we recover a well-known statement as special case of Theorem 1 — for two qubits, the set of local operations together with any entangling two-qubit gate form a universal set. Here, having access to all local gates implies that the two original systems, the separate qubits, are evolution operator controllable. The entangling gate has the same effect as the entangling control that we use to connect the systems. Note that already in the bipartite case, Theorem 1 is slightly more general: Substitute *qudits* for the qubits, perhaps of different dimensions $d_1 = 2^M$ and $d_2 = 2^N$. Since the study is restricted to qubit arrays, the dimension of the qudits is also restricted to powers of 2. The set of local unitary operators for two qubits $SU(2) \otimes SU(2)$ is replaced by $SU(d_1) \otimes SU(d_2)$. Every local gate in $SU(d_1) \otimes SU(d_2)$ can be implemented given the assumption of controllability on the separate partitions. Theorem 1 then states that given two controllable qudits with dimensions $d_1$ and $d_2$ and an entangling Hermitian operator $\hat{G}$, the set composed of local operations on the two qudits and the entangling rotation gates $\hat{R}_{\hat{G}}(\phi)$ form a universal set. This universality of the set of local gates plus an entangling gate on qudits of the same size is a lesser known fact in quantum information [35–37].

Once two controllable systems are connected by an entangling control (e.g. a tunable coupler), they can be understood as a controllable system in itself. This implies that we can connect it to a different controllable system using a new entangling control to build a larger yet controllable system. Note that the type of entangling operation between partitions is not relevant to prove controllability. In particular, the statement is true for any two-qubit tunable coupling as long as each of the qubits belongs to a different partition.

Naturally, having only one entangling operation between two partitions will limit the speed at which information (and in particular entanglement) can travel through the system. At the same time, a modular approach will also ease transpilation of quantum algo-

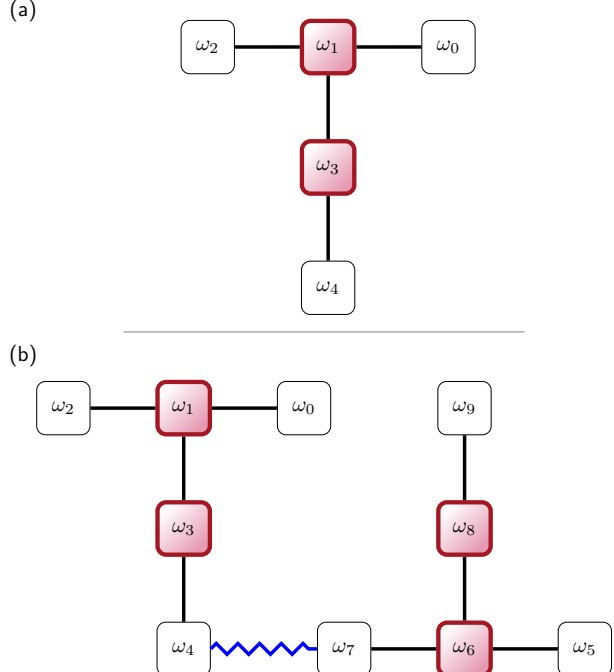

FIG. 1. (a): Controllable five-qubit system using just two controls [20]. Qubits with (without) local controls are shaded (blank) and static couplings are drawn as straight black lines. (b): Controllable system built up from two copies of the system in (a) with a tunable coupling (blue zig-zag connection).

rithms, making the use of partitions that are large enough to run certain subtasks a viable option. This type of architecture is also ideal for using each partition as sub-unit when parallelizing a more complex quantum algorithm. While these use cases would be specifically tailored to the modular design, the universality of the composite system ensures its use it for *any* kind of quantum circuit.

As an example, consider a T-shaped five-qubit system, inspired by the former five-qubit *ibm_quito* quantum processor by IBM (cf. https://quantum.cloud.ibm.com/docs/en/guides/retired-qpus), the controllability of which has been proven using a graph method [20]. When properly placed, two local controls are sufficient for the five-qubit system with fixed two-qubit couplings in the form of $\hat{H}^{(k,l)} = \hat{X}^{(k)}\hat{X}^{(l)} + \hat{Y}^{(k)}\hat{Y}^{(l)}$ to be evolution operator controllable, as illustrated also in Fig. 1(a). Its Hamiltonian is given by

$$\hat{H}^{5Q}(t) = -\sum_{j=0}^{4} \frac{\omega_j}{2}\hat{Z}^{(j)} + \sum_{\substack{(k,l)\in\{(0,1),(1,2),\\(1,3),(3,4)\}}} J_{k,l}\hat{H}^{(k,l)}$$
$$+ u_1(t)\hat{X}^{(1)} + u_2(t)\hat{X}^{(3)}, \quad (3)$$

where $u_{1/2}(t)$ represent the two local controls, $\omega_j$ are the local qubit frequencies and $J_{k,l}$ the coupling strengths.

Connecting two such five-qubit arrays by a tunable coupling, assumed to be of the form $\hat{X}^{(4)}\hat{X}^{(7)}$, acting

on the fourth and seventh qubit, the Hamiltonian of the composite system is written as

$$\hat{H}(t)^{2T_5} = \hat{H}^{5Q,\,A}(t)\otimes\mathbb{1}+\mathbb{1}\otimes\hat{H}^{5Q,\,B}(t)+w_{ent}(t)\hat{X}^{(4)}\hat{X}^{(7)}. \quad (4)$$

where the partial Hamiltonians $\hat{H}^{5Q,\,(A/B)}(t)$ are given by Eq. (3) (with the sole difference that the qubit frequencies $\omega_j^{(A/B)}$ and coupling strengths $J_{i,j}^{(A/B)}$ will differ). The controls for both partitions are also meant to be taken as independent. Evidently, qubit 4 and 7 belong to partitions $A$ and $B$, respectively. Note that the choice of specific qubits is fully arbitrary, and any other entangling control between qubits in the two subsystems would have done the trick.

Once we have constructed a system that is evolution operator controllable, like the one displayed in Fig. 1(b), we can add more controls without affecting controllability. On the contrary, this can be beneficial, allowing for faster unitary gates or generating the same Lie algebra using commutators of lower depth. As an example, consider the task of implementing an entangling operation on the zeroth and ninth qubit in Fig. 1(b). Intuitively, the fastest option is expected if a tunable coupling between those two qubits exists, whereas the two qubits are at a distance of seven (six static couplings plus the entangling control following the chain 0-1-3-4-7-6-8-9) with the setup of Fig. 1(b). This example illustrates a typical trade-off between connectivity and operation time: The quantum speed limit of gates [38–42] in large qubit arrays with minimal number of controls and couplings may grossly surpass the decoherence limit of the system.

## III. DESIGN OF LARGE QUBIT ARRAYS

Next we discuss how to use Theorem 1 to identify possible designs for large qubit arrays with fewer controls and couplings than those currently in use (knowing that the question of operation time will also have to be addressed). To showcase this, we take as basic layout one of the quantum processors from IBM and design a similar architecture using smaller QPUs that have already been shown to be controllable. This will allow us to identify local controls and couplings that can be removed while maintaining evolution operator controllability, i.e., the capability for universal computing.

The start point is a 127-qubit system with similar nearest-neighbor connectivity as shown in Fig. 2, but with local controls on every qubit and tunable two-qubit couplings on every shown connection. This two-dimensional layout is well suited for our approach, as it already has a regular pattern consisting mainly of three-by-five rectangles. The size of this device would make proving controllability using the graph test [20] or the dimensional expressivity test [21] extremely challenging.

The first building block, or "module", is a T-shaped

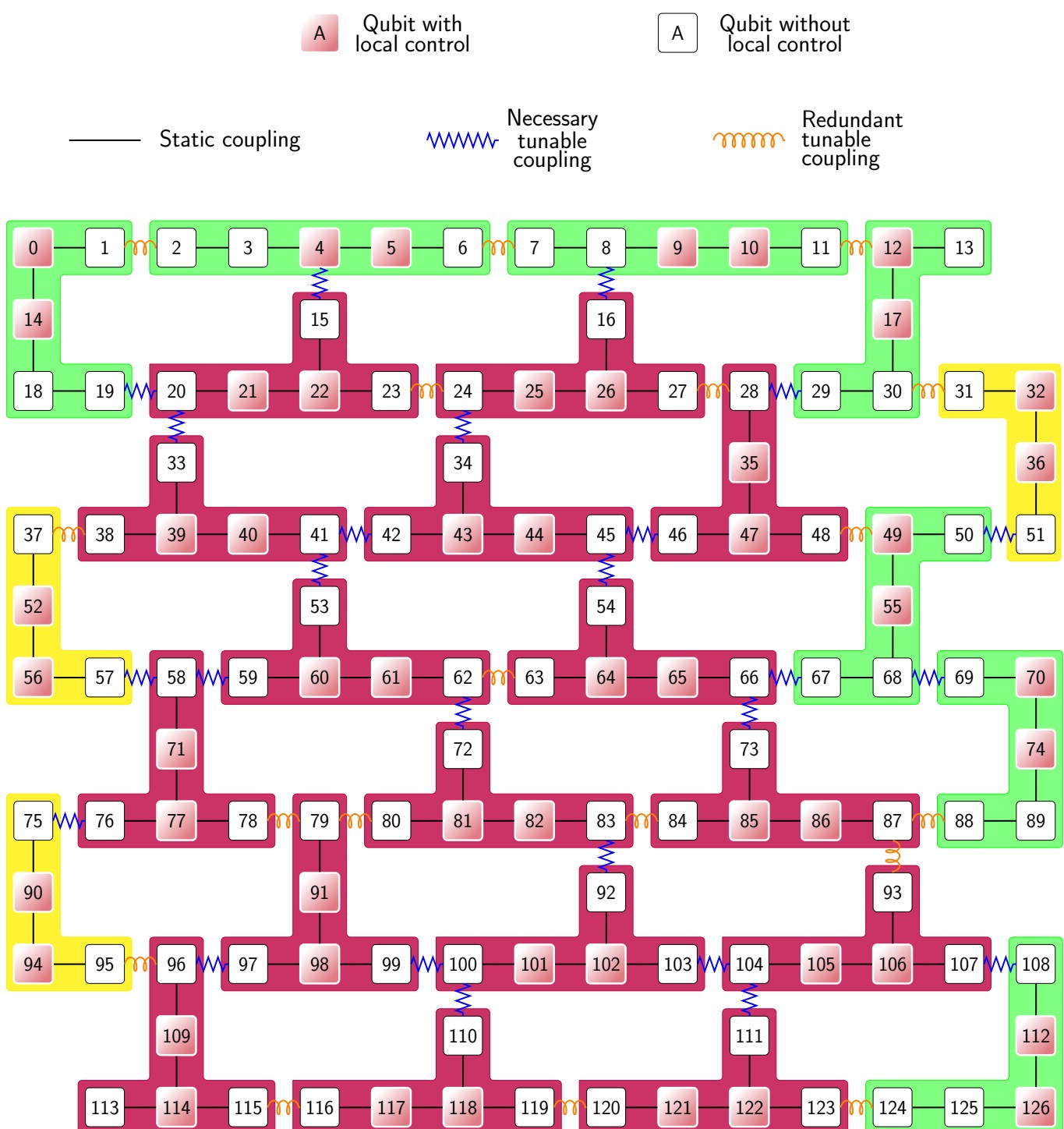

FIG. 2. Controllable qubit array starting from the connectivity of IBM's Eagle processor (cf. https://quantum.cloud.ibm.com/docs/en/guides/processor-types) but with a reduced number of controls and couplings, based on a decomposition into three different modules consisting of four, resp. five qubits (indicated by the yellow, green and red background color). Shaded qubits are equipped with local controls, while the blank ones are not. Blue zigzag lines are required tunable couplings, whereas coiled lines represent tunable couplings that are present in Eagle but can be removed without hampering controllability. The couplings within each module are static, as opposed to the other tunable ones. The Hamiltonians for the three modules are given in Eq. (5).

five-qubit system. Its Hamiltonian can be written as

$$\hat{H}_{5T}(t) = \sum_{j=0}^{4} \hat{H}_{1qubit}^{(j)} + \hat{H}_{control}^{(1)}(t) + \hat{H}_{control}^{(3)}(t) \quad (5a)$$
$$+ \hat{H}_{coup}^{(0,1)} + \hat{H}_{coup}^{(1,2)} + \hat{H}_{coup}^{(1,3)} + \hat{H}_{coup}^{(3,4)},$$

where the single-qubit Hamiltonian $\hat{H}_{1qubit}^{(j)}$, the control Hamiltonian $\hat{H}_{control}^{(j)}(t)$ and the static coupling Hamiltonian $\hat{H}_{coup}^{(j,k)}$ are

$$\hat{H}_{1qubit}^{(j)} = -\frac{\omega_j}{2}\hat{\sigma}_z^{(j)}, \qquad \hat{H}_{control}^{(j)}(t) = u_j(t)\hat{\sigma}_x^{(j)},$$
$$\hat{H}_{coup}^{(j,k)} = J_{j,k}\left(\hat{\sigma}_x^{(j)}\hat{\sigma}_x^{(k)} + \hat{\sigma}_y^{(j)}\hat{\sigma}_y^{(k)}\right)$$
$$(5b)$$

for qubit frequencies $\omega_j$, coupling strengths $J_{j,k}$ and controls $u_j(t)$. This configuration is based on an older five-qubit system by IBM, see also Fig. 1 (a). Two more building blocks provide a cover for the 127-qubit array. One is arbitrarily chosen to be a five-qubit system arranged in a line with Hamiltonian

$$\hat{H}_{5L}(t) = \sum_{j=0}^{4} \hat{H}_{qubit}^{(j)} + \hat{H}_{control}^{(1)}(t) + \hat{H}_{control}^{(2)}(t) \quad (5c)$$
$$+ \hat{H}_{coup}^{(0,1)} + \hat{H}_{coup}^{(1,2)} + \hat{H}_{coup}^{(2,3)} + \hat{H}_{coup}^{(3,4)},$$

with four static couplings and two local controls. For the other one, removing one qubit from Eq. (5c) defines a four-qubit line with similar features,

$$\hat{H}_{4L}(t) = \sum_{j=0}^{4} \hat{H}_{qubit}^{(j)} + \hat{H}_{control}^{(1)}(t) + \hat{H}_{control}^{(2)}(t) \quad (5d)$$
$$+ \hat{H}_{coup}^{(0,1)} + \hat{H}_{coup}^{(1,2)} + \hat{H}_{coup}^{(2,3)} + \hat{H}_{coup}^{(3,4)}.$$

Using copies of the systems with $\hat{H}_{5L}(t)$, $\hat{H}_{5L}(t)$ and $\hat{H}_{5L}(t)$, the complete 127-qubit device can be built up. To make the total system controllable, according to Theorem 1, each subsytem must be connected to another one using a tunable coupling. The result is shown in Figure 2, where some of the tunable couplings from the original layout have been "down-graded" to static couplings, effectively reducing the numbers of controls in the system, and some further ones have been removed altogether (the ones depicted with coiled orange lines) without losing controllability. The main achievement, however, is the reduction in the number of local controls to roughly two fifths of the original amount. This is obtained simply by reducing the number of controls in each module by finding controllable four- and five-qubit arrays with any desired controllability test. Comparing exact numbers, IBM's system is composed of 127 local controls and 144 tunable couplings; the alternative shown in Fig. 2 has only 52 local controls, 101 static couplings, and 25 tunable couplings (marked as necessary couplings). In

addition to these 25 couplings, there are a total of 18 extra tunable couplings which can be removed to lower the number of resources or left in place to speed up quantum information transfer. The modified system is a simplified version of the original one with fewer resources.

As a last note, we can use Theorem 1 also to prove that the original IBM system is controllable. To this end, simply recall two properties of controlled systems: (i) If a quantum system is controllable, the same system with more controls must be controllable, too. We have shown the system in Fig. 2 to be controllable with only the static couplings and the necessary tunable couplings (indicated by blue zigzag lines). If all local controls and the redundant tunable couplings are added back in, the system must be controllable, too. (ii) If a qubit array with static couplings is controllable, then the same system but with tunable couplings is also controllable, since the dynamical Lie algebra of the former is contained in the Lie algebra of the latter system. Hence the modified system in Fig. 2 with tunable instead of static couplings is controllable, and the system with local controls in every qubit and tunable couplings between any neighboring qubits must be controllable as well. In other words, we have proven that the original 127-qubit system with local controls on every qubit is evolution operator controllable, illustrating another possible application of Theorem 1.

## IV. SUMMARY

We have presented a method to assemble arbitrarily large controllable qubit arrays from smaller ones, proving rigorously that a single entangling two-qubit gate is sufficient to connect any pair of subsystems, irrespective of their specific layout, provided each subsystem is evolution operator controllable. Taking the IBM Eagle quantum processor as example, we have shown that evolution operator controllability can be guaranteed using far fewer couplings and controls than in the original system. The theorem thus provides a template for the modular design of QPUs, allocating controls and couplings based on their necessity for evolution operator controllability, a requirement for universal quantum computing.

While a smaller number of external controls frees up physical space on the chip and reduces calibration efforts, it comes at the price of longer operation times. In future work, it will therefore be interesting to compare the minimal evolution time, or quantum speed limit, of the total system with those of the subsystems from which it is built up. The rate at which information can be exchanged in the multipartite system will depend on various factors, including the maximum amplitude of the entangling control and the coupling distance between any two qubits in the total array. To mitigate a possibly higher quantum speed limit, more than one tunable coupling can be added to speed up the implementation of entangling unitary operations.

Another option, even if information transfer between

subsystems is slow, is to tailor the system architecture to run quantum algorithms that can be parallelized. Using each subsystem as a QPU, quantum circuits could be designed such that calculations are run in parallel on each subsystem and, at the end, all the information is merged before measuring. The parallelization of quantum algorithms is a current topic of discussion and has been hypothesized, for example, to allow for reduced circuit depths in some cases [43] or help with the implementation of decision diagrams [44].

In future work, it will also be interesting to extend the proof to systems that are not based on qubits. Throughout all lemmas, we have assumed two-level systems as elementary building blocks, allowing us to use tensor products of Pauli matrices as the basis for the dynamical Lie algebras. It thus needs to be shown that the same arguments can be made using e.g. the generalized skew-Hermitian Pauli matrices as a basis.

## ACKNOWLEDGMENTS

Financial support from the Einstein Research Foundation (Einstein Research Unit on Near-Term Quantum Devices) and the Deutsche Forschungsgemeinschaft project KO 2301/15-1, No. 505622963 is gratefully acknowledged. This work is supported with funds from the Ministry of Science, Research and Culture of the State of Brandenburg within the Centre for Quantum Technology and Applications (CQTA).

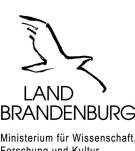

LAND
BRANDENBURG
Ministerium für Wissenschaft,
Forschung und Kultur

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

## Appendix A: Proof of the theorem

To prove Theorem 1, we use an analytical approach to prove controllability of the multipartite system using the Lie rank condition [19]. This section proves that when adding any two-qubit tunable coupling between the subsystems $\mathcal{A}$ and $\mathcal{B}$, the whole system $\mathcal{A} \otimes \mathcal{B}$ is controllable.

### 1. Notation and problem statement

Let $\mathfrak{A} \cong \mathfrak{su}(2^M)$ and $\mathfrak{B} \cong \mathfrak{su}(2^N)$ be the dynamical Lie algebras of two separate controllable qubit arrays $\mathcal{A}$ and $\mathcal{B}$, with $M$ and $N$ qubits respectively. The elements of the algebra $\mathfrak{A}$ of the first subsystem can be represented as linear combinations of tensor products of Pauli matrices $\hat{\sigma}_\alpha$ (with $\alpha \in \{0, 1, 2, 3\}$). We represent single Pauli matrices in $\mathfrak{A}$ by

$$\hat{\sigma}_\alpha^{(\mu)} := \mathbb{1}_2 \otimes ... \otimes \mathbb{1}_2 \otimes \underbrace{\hat{\sigma}_\alpha}_{\mu\text{-th position}} \otimes \mathbb{1}_2 \otimes ... \mathbb{1}_2,$$

with $\alpha \in [0, 1, 2, 3]$. Similarly, we represent the elements of $\mathfrak{B}$ following the same notation

$$\hat{\sigma}_j^{(n)} := \mathbb{1}_2 \otimes ... \otimes \mathbb{1}_2 \otimes \underbrace{\hat{\sigma}_j}_{n\text{-th position}} \otimes \mathbb{1}_2 \otimes ... \mathbb{1}_2,$$

with $j \in [0, 1, 2, 3]$. To avoid confusion we use Greek letters for the indices of elements belonging to $\mathfrak{A}$ and Latin ones for those in $\mathfrak{B}$.

Let us assume that the system $\mathcal{A}$ is described by the traceless Hamiltonian:

$$\hat{H}^\mathcal{A}(t) = \hat{A}_0 + \sum_{\rho=1}^{C_A} u_\rho(t)\hat{A}_\rho, \tag{A1}$$

where $\hat{A}_0$ is the time-independent drift, $u_\rho(t)$ are the real-valued controls, $\hat{A}_\rho$ the operators to which the controls are coupled and $C_A$ the number of controls in the system. The dynamical Lie algebra of the system is given by

$$\mathfrak{A} := Lie\left[\{i\hat{A}_\rho\}_{\rho=0}^{C_A}\right] \simeq \mathfrak{su}(2^M). \tag{A2}$$

Analogously, we assume the system $\mathcal{B}$ to be described by

$$\hat{H}^\mathcal{B}(t) = \hat{B}_0 + \sum_{j=1}^{C_B} v_j(t)\hat{B}_j, \tag{A3}$$

with a dynamical Lie algebra

$$\mathfrak{B} := Lie\left[\{i\hat{B}_j\}_{j=0}^{C_B}\right] \simeq \mathfrak{su}(2^N). \tag{A4}$$

Connecting two qubits from $\mathcal{A}$ and $\mathcal{B}$ by an entangling two-qubit coupling $w(t)\hat{H}_c^{\mu,n}$ (cf. Equation Eq. (2)) that can be treated as an independent control, the Hamiltonian of the $\mathcal{A} \otimes \mathcal{B}$ system is given by

$$\hat{H}^{\mathcal{AB}}(t) = \hat{H}^\mathcal{A}(t) \otimes \mathbb{1}_{2^N} + \mathbb{1}_{2^M} \otimes \hat{H}^\mathcal{B}(t) + w(t)\hat{H}_c^{\mu,n}$$

$$= \hat{A}_0 \otimes \mathbb{1}_{2^N} + \mathbb{1}_{2^M} \otimes \hat{B}_0 + \sum_{\rho=1}^{C_A} u_\rho(t)\hat{A}_\rho \otimes \mathbb{1}_{2^N} + \sum_{j=1}^{C_B} v_j(t)\mathbb{1}_{2^M} \otimes \hat{B}_j + w(t)\hat{H}_c^{\mu,n}. \tag{A5}$$

Note that the drift of $\hat{H}^{\mathcal{AB}}$, $\hat{H}_0^{\mathcal{AB}} = \hat{A}_0 \otimes \mathbb{1}_{2^N} + \mathbb{1}_{2^M} \otimes \hat{B}_0$, encompasses the contributions of the drifts of $\mathcal{A}$ and $\mathcal{B}$. The dynamical Lie algebra of the bipartite system is given by

$$\mathcal{L}_{\mathcal{A} \otimes \mathcal{B}} := Lie\left[i\hat{H}_0^{\mathcal{AB}}, \{i\hat{A}_\rho \otimes \mathbb{1}_{2^N}\}_{\rho=1}^{C_A}, \{i\mathbb{1}_{2^M} \otimes \hat{B}_j\}_{j=1}^{C_B}, i\hat{H}_c^{\mu,n}\right], \tag{A6}$$

which is contained in $\mathcal{L}_{\mathcal{A} \otimes \mathcal{B}} \subseteq \mathfrak{su}(2^{M+N})$. The different operators in Eq. (A6) include, in order of appearance, the drift of the total system, the local controls on the partition $\mathcal{A}$, the local controls on the partition $\mathcal{B}$, and the entangling control. Here we prove that for two operator controllable systems $\mathcal{A}$ and $\mathcal{B}$, described by Eq. (A1) and Eq. (A3), respectively, the bipartite system $\mathcal{A} \otimes \mathcal{B}$ with the two-qubit control $w(t) \hat{H}_c^{\mu,n}(t)$ is also operator controllable. Mathematically, this is equivalent to

$$
\begin{cases} \mathfrak{A} = \mathfrak{su}(2^M) \\ \mathfrak{B} = \mathfrak{su}(2^N) \end{cases} \Rightarrow \quad \mathcal{L}_{\mathcal{A} \otimes \mathcal{B}} = \mathfrak{su}(2^{M+N}) \tag{A7}
$$

for every $\hat{H}_c^{\mu,n}$ as in Eq. (2).

## 2. Operations with tensor products of Pauli matrices

Most of the needed calculations to generate the dynamical Lie algebra of the total system described by Eq. (A6) involve commutators of the entangling control $\hat{H}_c^{\mu,n}$ with local operators of either subsystem. It is therefore important to understand what elements can be generated using only linear combinations of commutators of local operators. Controllability of the total system corresponds to obtaining every element of the total Lie algebra. To ensure this, it is enough to show that a complete basis of the Lie algebra can be generated. For the case of the maximal algebra of $\mathcal{A} \otimes \mathcal{B}$, $\mathfrak{su}(2^{N+M})$, a possible basis is

$$
\mathfrak{su}(2^{N+M}) = \text{span} \left\{ i \, \hat{\sigma}_{\alpha_1} \otimes \hat{\sigma}_{\alpha_2} \otimes \cdots \otimes \hat{\sigma}_{\alpha_M} \otimes \hat{\sigma}_{j_1} \otimes \hat{\sigma}_{j_2} \otimes \cdots \otimes \hat{\sigma}_{j_N} \right\}_{j_k, \alpha_\beta = 0}^3 \tag{A8}
$$

with at least one nonzero subindex $j_k$ or $\alpha_\beta$ in every element. All of the elements from the basis in Eq. (A8) present the same structure with different subindices. If we start from an element in the basis and define operations to change the Pauli indices to other possible values, we can generate a basis of $\mathfrak{su}(2^{N+M})$ and prove controllability.

There are three qualitatively different possible index changes: Changing a nonzero index to a nonzero index, turning a nonzero index to zero or setting a zero index to a nonzero index. Here we introduce the associated three operations using local commutators on the $\mathfrak{B}$ subsystem. From here on, all subscripts $b$ of the Pauli operators $\hat{\sigma}_b^a$ are assumed to be nonzero unless explicitly stated otherwise. For simplicity, we also use a cyclic notation in the Pauli indices $\{1, 2, 3\}$, e.g. $\hat{\sigma}_{3+1} = \hat{\sigma}_4 := \hat{\sigma}_1$. We start with an operation that allows us to change an index $j \in \{1, 2, 3\}$ of a Pauli matrix $\hat{\sigma}_j^{(n)}$ to $j + 1$ for tensor products of Pauli matrices in the subsystem $\mathcal{B}$:

$$
f_{cyc}^{(n)} \left( i \hat{\sigma}_j^{(n)} \right) := -\frac{1}{2} [i \hat{\sigma}_{j+2}^{(n)}, \, i \hat{\sigma}_j^{(n)}] = i \hat{\sigma}_{j+1}^{(n)}. \tag{A9a}
$$

Note that $f_{cyc}^{(n)}(\cdot)$ is defined using only commutators of elements in the Lie algebra $\mathfrak{B}$. Analogous operations can be defined for other values of $1 \leq n \leq N$. With it we can cycle through all the nonzero indices of a Pauli matrix. This operation has some interesting properties. First, if we have a skew-Hermitian operator $i \hat{\sigma}_j^{(n)} \hat{\sigma}_j^{(m)}$ with $m \neq n$, then $f_{cyc}^{(n)}(i \hat{\sigma}_k^{(n)} \hat{\sigma}_k^{(m)}) := f_{cyc}^{(n)}(i \hat{\sigma}_j^{(n)}) \hat{\sigma}_k^{(m)}$. In other words, if we have a tensor product of Pauli matrices, $f_{cyc}^{(n)}$ does not change the indices of Pauli matrices in a position $m \neq n$. Second, for Hermitian operators $\hat{A}$ acting on the first system $\mathcal{A}$, the definition can be naturally extended to $f_{cyc}^{(n)}(i \hat{A} \otimes \hat{\sigma}_j^{(n)}) := \hat{A} \otimes f_{cyc}^{(n)}(i \hat{\sigma}_j^{(n)})$. This operation will be relevant to compute the Lie algebra of the multipartite system. If $i \hat{A} \otimes \hat{\sigma}_j^{(n)} \in \mathcal{L}_{\mathcal{A} \otimes \mathcal{B}}$ and for the local operator $i \mathbb{1}_{2^M} \otimes \hat{\sigma}_{j+2}^{(n)} \in \mathcal{L}_{\mathcal{A} \otimes \mathcal{B}}$, then $\hat{A} \otimes f_{cyc}^{(n)}(i \hat{\sigma}_j^{(n)}) \in \mathcal{L}_{\mathcal{A} \otimes \mathcal{B}}$. The second hypothesis is trivially fulfilled if $\mathbb{1}_{2^M} \otimes \mathfrak{su}(N) \in \mathcal{L}_{\mathcal{A} \otimes \mathcal{B}}$, which is something that still has to be formally proven.

For the second operation we define a function using commutators of operators in $\mathcal{B}$ to turn a Pauli index from zero to $j > 0$ at a position $n$. To achieve this it is imperative to have at least another Pauli matrix $\hat{\sigma}_k^{(m)}$ at a position $m \neq n$. The operation is given by

$$
f_{gen\,k}^{(n,m)} \left( i \hat{\sigma}_0^{(n)} \hat{\sigma}_k^{(m)} \right) := -\frac{1}{4} \left[ i \hat{\sigma}_0^{(n)} \hat{\sigma}_{k+1}^{(m)}, \, \left[ i \hat{\sigma}_j^{(n)} \hat{\sigma}_{k+1}^{(m)}, \, i \hat{\sigma}_0^{(n)} \hat{\sigma}_k^{(m)} \right] \right] = i \hat{\sigma}_j^{(n)} \hat{\sigma}_k^{(m)}. \tag{A9b}
$$

Analogous operations can be defined for other values of $1 \leq n, m \leq N$ and $j \in \{1, 2, 3\}$. Similarly as before, the function $f_{gen\,k}^{(n,m)}$ can be extended to other operators $i \mathcal{A} \otimes \hat{\sigma}_0^{(n)} \hat{\sigma}_k^{(m)}$ acting on the total Hilbert space. By definition, this operation is not affected by any other Pauli matrices at any position other than $m$ and $n$.

Finally, the third operation removes a Pauli matrix from the tensor product, i.e. it takes an element $\hat{\sigma}_j^{(n)}$ and sets it to $\hat{\sigma}_0^{(n)}$. To do so it is necessary to have another Pauli matrix $\hat{\sigma}_k^{(m)}$ in the tensor product. The operation is defined as

$$f_{rem}^{(n,m)}\left(i\hat{\sigma}_j^{(n)}\hat{\sigma}_k^{(m)}\right) := -\frac{1}{4}\left[i\hat{\sigma}_0^{(n)}\hat{\sigma}_{k+1}^{(m)},\ \left[i\hat{\sigma}_j^{(n)}\hat{\sigma}_{k+1}^{(m)},\ i\hat{\sigma}_j^{(n)}\hat{\sigma}_k^{(m)}\right]\right] = \hat{\sigma}_0^{(n)}\hat{\sigma}_k^{(m)}, \tag{A9c}$$

where similar functions can be defined for other positions $n, m$ and indices $j, k$. The same properties described for the operations $f_{cyc}^{(n)}$ and $f_{gen\,k}^{(n,m)}$ apply to $f_{rem}^{(n,m)}$.

We can define operations $f_{cyc}^{(\rho)}$, $f_{gen\,k}^{(\rho,\tau)}$ and $f_{rem}^{(\rho,\tau)}$ analogous to Eq. (A9) acting on the subsystem $\mathcal{A}$ instead of the subsystem $\mathcal{B}$. With this set of operations we can transform any Pauli matrix tensor product (cf. Eq. (A8)) with at least one Pauli matrix acting on $\mathcal{A}$ and one acting on $\mathcal{B}$ and transform it into any other tensor product of Pauli matrices that is entangling between $\mathcal{A}$ and $\mathcal{B}$. In other words, if we have access to all the different local operations then we can transform any non-local tensor product of Pauli matrices into any other non-local tensor product of Pauli matrices. This will be key in turning the entangling tunable coupling into other elements in the total Lie algebra by only using local operations, i.e., extending local controllability of the subsystems to the total system $\mathcal{A} \otimes \mathcal{B}$.

## 3. Preliminaries

In this subsection we present and prove a collection of lemmas that are necessary for the final theorem. Together, they focus on proving that the dynamical Lie algebras $\mathfrak{A} \simeq \mathfrak{su}(2^M)$, $\mathfrak{su}(2^M) \simeq \mathfrak{B}$ of the initial subsystems belong to the dynamical Lie algebra $\mathcal{L}_{\mathcal{A}\otimes\mathcal{B}}$ of the total system, i.e. that $\mathfrak{A} \otimes \mathbb{1}_{2^N} \subseteq \mathcal{L}_{\mathcal{A}\otimes\mathcal{B}}$, and $\mathbb{1}_{2^M} \otimes \mathfrak{B} \subseteq \mathcal{L}_{\mathcal{A}\otimes\mathcal{B}}$.

Bear in mind that this result is not immediate. Eq. (A2) (respectively Eq. (A4)) shows that all elements in $\{i\hat{A}_\rho\}_{\rho=0}^{C_A}$ (resp. $\{i\hat{B}_\rho\}_{j=0}^{C_B}$) are necessary to generate the Lie algebra $\mathfrak{A}$ (resp. $\mathfrak{B}$) in the general case. Looking at the elements that generate $\mathcal{L}_{\mathcal{A}\otimes\mathcal{B}}$ in Eq. (A6), one can see that neither $i\hat{A}_0$ nor $i\hat{B}_0$ appear directly among the elements. They appear, however, in the form of $\hat{H}_0^{\mathcal{A}\mathcal{B}}$. Indeed they are both time independent elements, which means that a priori we cannot control them independently. Therefore it is necessary to prove that, by using commutators with other elements and their linear combinations, it is possible to generate the algebras $\mathfrak{A} \otimes \mathbb{1}_{2^N}$ and $\mathbb{1}_{2^M} \otimes \mathfrak{B}$.

First, we present two lemmas that combined show that $\mathfrak{A} \otimes \mathbb{1}_{2^N}$ can be generated using the elements in Eq. (A6). Intuitively it should be possible to separate them, as they act on different subspaces. Here that suspicion is formally proven.

**Lemma 2.** *Let* $\{\mathbf{v}_i\}_{i=0}^m$ *be a set of vectors in a vector space* $V_A$. *Let* $\mathbf{w} \in V_B$ *be another vector on a vector space* $V_B$ *such that* $V_A \cap V_B = \{\mathbf{0}\}$. *If the set* $\{\mathbf{v}_i\}_{i=1}^m$ *is linearly independent, then the set* $\{\mathbf{v}_0 + \mathbf{w}\} \cup \{\mathbf{v}_i\}_{i=1}^m$ *is also linearly independent.*

*Proof.* To prove that the second set is linearly independent we simply must show that for the following equality to hold,

$$c_0\left(\mathbf{v}_0 + \mathbf{w}\right) + \sum_{i=1}^m c_i\mathbf{v}_i = 0,$$

all the coefficients $c_j$ (with $0 \leq j \leq m$) must be zero. Since $\mathbf{w} \perp \mathbf{v}_i \,\forall j \in \{1, 2, ..., m\}$ it is evident that $c_0 = 0$. This leaves us with the equation

$$\sum_{i=1}^m c_i\mathbf{v}_i = 0.$$

If the set $\{\mathbf{v}_i\}_{i=1}^m$ is linearly independent, then $c_i = 0 \,\forall j \in \{1, 2, ..., m\}$, which means that the elements in the set $\{\mathbf{v}_0 + \mathbf{w}\} \cup \{\mathbf{v}_i\}_{i=1}^m$ are also linearly independent. $\qquad\square$

The next lemma includes a mention to the generation of a vector basis of a Lie algebra $\mathfrak{A}$ based on a set of elements $\{A_1, A_2, ...A_n\}$. Here is a short description of one of the possible methods, extracted from described in Chapter 3 of [19]:

1. Orthonormalise the elements of the initial set $\{A_1, A_2, ...A_n\}$ into a maximal set of linearly independent orthonormal elements $\{\tilde{A}_1, \tilde{A}_2, ...\tilde{A}_m\}$ (e.g. by running the Gram-Schmidt algorithm).

2. Define $\{\tilde{A}_1, \tilde{A}_2, ...\tilde{A}_m\}$ as the elements of depth 0.

3. $p \leftarrow 1$.

4. Iterate over the next steps:

   (a) Compute the Lie brackets $C_{i,j} := [\tilde{B}_j, \tilde{A}_i]$ where $\tilde{A}_i$ are the elements of depth 0 and $\tilde{B}_j$ the elements of depth $p - 1$. The vectors $C_{i,j}$ are potential candidates for elements of depth $p$.

   (b) Orthonormalise the set of vectors $\{C_{i,j}\}$ with respect to the set of elements of depth less or equal than $p - 1$.

   (c) The new nonzero vectors $\tilde{C}_k$ are considered the elements of depth $p$ from now on.

   (d) Stop the algorithm if the set of elements of depth $p$ is empty (i.e. no new linearly independent vector was found through Lie brackets).

   (e) $p \leftarrow p + 1$

**Lemma 3.** *Let $\{i\hat{A}_\rho\}_{\rho=0}^{C_A} \subset \mathfrak{su}(2^M)$ be a set of linearly independent traceless skew-Hermitian operators that generate the algebra $\mathfrak{A} := Lie\left[\{i\hat{A}_\rho\}_{\rho=0}^{C_A}\right]$. Let $i\hat{B}_0 \in \mathfrak{su}(2^N)$ be a nonzero operator such that $[i\hat{A}_\rho \otimes \mathbb{1}_{2^N} i\mathbb{1}_{2^M} \otimes \hat{B}_0] = 0$ for every $0 \leq \rho \leq C_A$. Then the following implication holds true:*

$$\text{If } \mathfrak{A} \cong \mathfrak{su}(2^M) \text{ and } \mathfrak{A}' := Lie\left[i\hat{A}_0 \otimes \mathbb{1}_{2^N} + i\mathbb{1}_{2^M} \otimes \hat{B}_0, \{i\hat{A}_\rho \otimes \mathbb{1}_{2^N}\}_{\rho=1}^{C_A}\right]$$
$$\text{then } \mathfrak{A}' \cong \mathfrak{su}(2^M) \otimes \mathbb{1}_{2^N} \bigoplus i\mathbb{1}_{2^M} \otimes \hat{B}_0. \tag{A10}$$

*Proof.* Assume that the hypothesis $\mathfrak{A} \cong \mathfrak{su}(2^M)$ is true. Let $\{\hat{G}_\rho\}_{\rho=0}^{2^{2M}-2}$ be the basis of $\mathfrak{A}$ spanned by $\{i\hat{A}_\rho\}_{\rho=0}^{C_A}$ generated by the previously described procedure. We are going to show that $\hat{G}'_\rho := \hat{G}_\rho \otimes \mathbb{1}_{2^N} \in \mathfrak{A}'$ for every $1 \leq \rho \leq 2^{2M} - 2$. Since $[i\hat{A}_\rho \otimes \mathbb{1}_{2^N}, i\mathbb{1}_{2^M} \otimes \hat{B}_0] = 0$ for every $\rho$, for every $\hat{G}_\rho \in \mathfrak{A}$ of depth $p \geq 1$ we can generate the associated $\hat{G}'_\rho \in \mathfrak{A}'$. Additionally, for every element $\hat{G}_\rho$ of depth $p = 0$ and index $\rho > 1$, $\hat{G}'_\rho = \hat{A}_\rho \otimes \mathbb{1}_{2^N} \in \mathfrak{A}'$ by definition. By virtue of Lemma 2, since $\{\hat{G}'_\rho\}_{\rho=0}^{2^{2M}-2}$ is a linearly independent set, the set $\{i\hat{A}_0 \otimes \mathbb{1}_{2^N} + i\mathbb{1}_{2^M} \otimes \hat{B}_0\} \cup \{\hat{G}'_\rho\}_{\rho=1}^{2^{2M}-2} \subset \mathfrak{A}'$ must me linearly independent as well.

We have found a $(2^{2M} - 1)$-dimensional set of linearly independent terms in $\mathfrak{A}'$. Given that $\mathfrak{A}' \subseteq \mathfrak{su}(2^M) \otimes \mathbb{1}_{2^N} \bigoplus i\mathbb{1}_{2^M} \otimes \hat{B}_0$, then $2^{2M} - 1 \leq \dim(\mathfrak{A}') \leq 2^{2M}$.

Let us assume that $\dim(\mathfrak{A}') = 2^{2M} - 1$ to see that we reach a contradiction. The algebra

$$\mathfrak{G} := Lie\left[\{\hat{G}_\rho\}_{\rho=1}^{2^{2M}-2}\right] \tag{A11}$$

has dimension $\dim(\mathfrak{G}) = 2^{2M} - 2$. By definition of $\hat{G}_\rho$, the set $\{i\hat{A}_0, \hat{G}_1, ...\hat{G}_{2^{2M}-2}\}$ is a basis of $\mathfrak{A} \cong \mathfrak{su}(2^M)$. If $\dim(\mathfrak{A}') = 2^{2M} - 1$ then

$$[i\hat{A}_0 \otimes \mathbb{1}_{2^N} + i\mathbb{1}_{2^M} \otimes \hat{B}_0, \mathfrak{G} \otimes \mathbb{1}_{2^N}] = [i\hat{A}_0 \otimes \mathbb{1}_{2^N}, \mathfrak{G} \otimes \mathbb{1}_{2^N}] \subseteq \mathfrak{G} \otimes \mathbb{1}_{2^N}, \tag{A12}$$

which makes $\mathfrak{G}$ an ideal of $\mathfrak{su}(2^M)$ of dimension $2^{2M} - 2$. As a reminder, an ideal in a Lie algebra $\mathcal{L}$ is a vector subspace $\mathcal{I}$ so that $[\mathcal{L}, \mathcal{I}] \subseteq \mathcal{I}$. But this is not possible, since the special unitary algebras $\mathfrak{su}(n)$ are simple (i.e. they do not contain any non-trivial ideal). Thus $\dim(\mathfrak{G}) = 2^{2M} - 1$ and $\dim(\mathfrak{A}') = 2^{2M}$, proving the implication of this lemma. □

The last needed lemma shows how to generate the total algebra $\mathcal{L}_{\mathcal{A} \otimes \mathcal{B}}$ using the entangling control $\hat{H}_c^{\mu,n}$ and the two local dynamical Lie algebras. In other words, this lemma presents a method to obtain all the remaining entangling elements in the total algebra using only linear combinations of commutators.

**Lemma 4.** *Let $\mathfrak{A} \cong \mathfrak{su}(2^M)$ and $\mathfrak{B} \cong \mathfrak{su}(2^N)$. Let*

$$\hat{H}_c^{\mu,n} := \sum_{\alpha,j \in [1,2,3]} c_{\alpha,j} \hat{\sigma}_\alpha^{(\mu)} \otimes \hat{\sigma}_j^{(n)} \quad \in \mathfrak{A} \otimes \mathfrak{B} \tag{A13}$$

*be a nonzero Hermitian operator with $1 \leq \mu \leq M$, $1 \leq n \leq N$, $c_{\alpha,j} \in \mathbb{R}$, $i\hat{\sigma}_\alpha^{(\mu)} \in \mathfrak{A}$ and $i\hat{\sigma}_j^{(n)} \in \mathfrak{B}$. Then*

$$\mathcal{L} := Lie\left[\mathfrak{A} \otimes \mathbb{1}_{2^N}, \ \mathbb{1}_{2^M} \otimes \mathfrak{B}, \ i\hat{H}_c^{\mu,n}\right] \cong \mathfrak{su}(2^{M+N}) \tag{A14}$$

*Proof.* We use tensor products of Pauli matrices for the bases of the algebras $\mathfrak{su}(2^M)$, $\mathfrak{su}(2^M)$ and $\mathfrak{su}(2^{M+N})$.

Without loss of generality, we can assume the two qubit coupling to be of the form $\hat{H}_c^{\mu,n} = c_{\alpha,j}\hat{\sigma}_\alpha^{(\mu)} \otimes \hat{\sigma}_j^{(n)}$, with only one nonzero coefficient $c_{\alpha,j}$ for some fixed $\alpha, j \in [1,2,3]$. Indeed, assume that $\hat{H}_c^{\mu,n}$ has a nonzero contribution $\hat{\sigma}_3^{(\mu)} \otimes \hat{\sigma}_3^{(n)}$, i.e., $c_{3,3} \neq 0$. We can isolate that term by using commutators of $\hat{H}_c^{\mu,n}$ with elements of $\mathfrak{A} \otimes \mathbb{1}_{2^N}$ and $\mathbb{1}_{2^M} \otimes \mathfrak{B}$. Given the usual commutation relations we compute the following element:

$$\left[i\hat{\sigma}_3^{(\mu)} \otimes \mathbb{1}_{2^N},\, \left[i\hat{\sigma}_1^{(\mu)} \otimes \mathbb{1}_{2^N},\, i\sum_{\alpha,j=1}^{3} c_{\alpha,j}\hat{\sigma}_\alpha^{(\mu)} \otimes \hat{\sigma}_j^{(n)}\right]\right] =$$

$$= \left[i\hat{\sigma}_3^{(\mu)} \otimes \mathbb{1}_{2^N},\, 2i\sum_{j=1}^{3} c_{3,j}\hat{\sigma}_2^{(\mu)} \otimes \hat{\sigma}_j^{(n)} - 2i\sum_{k=1}^{3} c_{2,k}\hat{\sigma}_3^{(\mu)} \otimes \hat{\sigma}_k^{(n)}\right]$$

$$= 4i\sum_{j=1}^{3} c_{3,j}\hat{\sigma}_1^{(\mu)} \otimes \hat{\sigma}_j^{(n)} \quad \in \mathcal{L}, \tag{A15}$$

where we have eliminated all coefficients with $\alpha \neq 3$. Similarly,

$$\left[i\hat{\sigma}_3^{(\mu)} \otimes \mathbb{1}_{2^N},\, \left[i\hat{\sigma}_1^{(\mu)} \otimes \mathbb{1}_{2^N},\, 4i\sum_{j=1}^{3} c_{3,j}c_{3,j}\hat{\sigma}_1^{(\mu)}\hat{\sigma}_j^{(n)}\right]\right] = 16i\,c_{3,3}\,\hat{\sigma}_1^{(\mu)}\hat{\sigma}_1^{(n)} \quad \in \mathcal{L}. \tag{A16}$$

Therefore, using commutators we were able to obtain a single tensor product of Pauli matrices between algebras, $\hat{\sigma}_\alpha^{(\mu)} \otimes \hat{\sigma}_j^{(n)}$ (with $\alpha = j = 1$ in the previous example). For any two-qubit operator we can isolate one term and assume it to be in the form $\hat{\sigma}_\alpha^{(\mu)} \otimes \hat{\sigma}_j^{(n)}$. If

$$Lie\left[\mathfrak{A} \otimes \mathbb{1}_{2^N},\, \mathbb{1}_{2^M} \otimes \mathfrak{B},\, i\hat{\sigma}_\alpha^{(\mu)} \otimes \hat{\sigma}_j^{(n)}\right] \cong \mathfrak{su}(2^{M+N}) \tag{A17}$$

then Eq. (A14) is true, since

$$Lie\left[\mathfrak{A} \otimes \mathbb{1}_{2^N},\, \mathbb{1}_{2^M} \otimes \mathfrak{B},\, i\hat{\sigma}_\alpha^{(\mu)} \otimes \hat{\sigma}_j^{(n)}\right] \subseteq Lie\left[\mathfrak{A} \otimes \mathbb{1}_{2^N},\, \mathbb{1}_{2^M} \otimes \mathfrak{B},\, i\hat{H}_c^{\mu,n}\right]. \tag{A18}$$

To prove Eq. (A17), we show that we can obtain every element

$$i\left(\hat{\sigma}_{\alpha_1}^{(1)}...\hat{\sigma}_{\alpha_M}^{(M)}\right) \otimes \left(\hat{\sigma}_j^{(1)}...\hat{\sigma}_{j^N}^{(N)}\right) \qquad \forall \alpha_\beta, j_k \in [0,1,2,3], \tag{A19}$$

with at least one $\alpha_\beta$ or $j_k$ nonzero. These elements form a basis of $\mathfrak{su}(2^{M+N})$. Note that the elements where $\alpha_\beta = 0$ with $1 \leq \beta \leq M$ already belong to $\mathbb{1}_{2^M} \otimes \mathfrak{B}$, whereas the terms with $j_k = 0$ for every $0 \leq k \leq N$ belong to $\mathfrak{A} \otimes \mathbb{1}_{2^N}$. Therefore, we only have to prove it for the cases where at least one $\alpha_\beta$ and at least one $j_k$ are nonzero. Starting with the two qubit coupling $\hat{H}_c^{\mu,n} = c_{\alpha,j}\hat{\sigma}_\alpha^{(\mu)} \otimes \hat{\sigma}_j^{(n)}$ and using the operations $f_{cyc}^{(n)}$, $f_{gen\,k}^{(n,m)}$, $f_{rem}^{(n,m)}$, $f_{cyc}^{(\rho)}$, $f_{gen\,k}^{(\rho,\tau)}$ and $f_{rem}^{(\rho,\tau)}$ (for every possible value of $n$, $m$, $\rho$ and $\tau$) defined in Section A 2 we can obtain any non-local tensor product of Pauli matrices using only commutators of $\hat{H}_c^{\mu,n}$ and local operators. Therefore, we can generate a basis of $\mathfrak{su}(2^{M+N})$ and the proof concludes.

$\square$

With these three lemmas we have all the necessary intermediate results to prove the main idea of this work, which is properly shown and discussed in the following subsection.

### 4. Controllability of two controllable qubit arrays coupled via a two-qubit control

We finally can prove that the system formed by two controllable qubit arrays and a two-qubit tunable coupling that connects both is also operator controllable and hence a suitable candidate for universal quantum computing. The condition that must be satisfied had already been summarized in Eq. (A7).

*Proof of Theorem 1.* The Hamiltonians of systems $\mathcal{A}$ and $\mathcal{B}$ can be written as Equations (A1) and (A3), respectively. Thus the Hamiltonian of the bipartite system composed of subsystem $\mathcal{A}$, subsystem $\mathcal{B}$ and the tunable coupling $\hat{H}_c^{\mu,n}$ (cf. Equation Eq. (2)) follows the form of Equation Eq. (A5).

A simple calculation of the dynamical Lie algebra from Equation (A6) yields

$$
\begin{aligned}
\mathcal{L}_{\mathcal{A}\otimes\mathcal{B}} &= Lie\left[1,\ \{i\hat{A}_\rho \otimes \mathbb{1}_{2^N}\}_{\rho=1}^{C_A},\ \{i\mathbb{1}_{2^M} \otimes \hat{B}_j\}_{j=1}^{C_B},\ i\hat{H}_c^{\mu,n}\right] = \\
&= Lie\left[Lie\left[i\hat{H}_0^{\mathcal{AB}},\ \{i\hat{A}_\rho \otimes \mathbb{1}_{2^N}\}_{\rho=1}^{C_A}\right],\ \{i\mathbb{1}_{2^M} \otimes \hat{B}_j\}_{j=1}^{C_B},\ i\hat{H}_c^{\mu,n}\right] \overset{\text{Lemma 3}}{=} \\
&= Lie\left[\mathfrak{A} \otimes \mathbb{1}_{2^N} \bigoplus i\mathbb{1}_{2^M} \otimes \hat{B}_0,\ \{i\mathbb{1}_{2^M} \otimes \hat{B}_j\}_{j=1}^{C_B},\ i\hat{H}_c^{\mu,n}\right] = \\
&= Lie\left[\mathfrak{A} \otimes \mathbb{1}_{2^N},\ i\mathbb{1}_{2^M} \otimes \hat{B}_0,\ \{i\mathbb{1}_{2^M} \otimes \hat{B}_j\}_{j=1}^{C_B},\ i\hat{H}_c^{\mu,n}\right] = \\
&= Lie\left[\mathfrak{A} \otimes \mathbb{1}_{2^N},\ Lie\left[i\mathbb{1}_{2^M} \otimes \hat{B}_0,\ \{i\mathbb{1}_{2^M} \otimes \hat{B}_j\}_{j=1}^{C_B}\right],\ i\hat{H}_c^{\mu,n}\right] = \\
&= Lie\left[\mathfrak{A} \otimes \mathbb{1}_{2^N},\ \mathbb{1}_{2^M} \otimes \mathfrak{B},\ i\hat{H}_c^{\mu,n}\right] \overset{\mathcal{A},\mathcal{B}\ \text{operator controllable}}{=} \\
&= Lie\left[\mathfrak{su}(2^M) \otimes \mathbb{1}_{2^N},\ \mathbb{1}_{2^M} \otimes \mathfrak{su}(2^N),\ i\hat{H}_c^{\mu,n}\right] \overset{\text{Lemma 4}}{=} \\
&= \mathfrak{su}\left(2^{M+N}\right).
\end{aligned}
\tag{A20}
$$

Therefore the bipartite system is operator controllable for any entangling two-qubit coupling. $\qquad\square$

As a final note and without an explicit proof, this result can also be extended to any type of control that is entangling between the partitions $\mathcal{A}$ and $\mathcal{B}$. The main idea behind it is that the same process by which $i\hat{H}_0^{\mathcal{AB}}$ was reduced to a single product $c_{\alpha,j}\hat{\sigma}_\alpha^{(\mu)} \otimes \hat{\sigma}_j^{(n)}$ can be analogously defined for any type of entangling coupling.

At first, this may seem like an unnecessary remark, as two-qubit couplings are often more easily implemented than interactions between three or more qubits. But it opens up a range of possibilities that are not constricted by this hypothesis. For example, it includes global controls that operate simultaneously on every qubit, like electromagnetic fields acting on trapped ions or on NV centers. Furthermore, using entangling controls that affect more than two qubits may increase the ratio at which information is shared between the two subsystems, depending on the architecture of the remaining couplings and controls. This generality offers a great freedom when using controllable subsystems as building blocks of a larger, controllable system.