# Peer review of "Scalable modular architecture for universal quantum computation"

_SciPost Physics_

## Round 2 · Referee Report · Anonymous (Referee 2) · 2025-11-4

Strengths

The manuscript presents a clear and elegant theoretical result: two controllable qubit arrays, when linked by a single entangling two-qubit gate, form a composite system that remains evolution-operator controllable. This theorem establishes a scalable and modular design principle for quantum processing units (QPUs), enabling universal quantum computation with reduced numbers of controls and couplings.

The paper is well-written and succeeds in reformulating a central question in quantum control theory: how controllability can be extended from smaller subsystems to larger architectures, in a way that is both rigorous and practically relevant. The Lie-algebraic proof provided in Appendix A is logically structured, with clear notation and sound reasoning. The connection made between abstract controllability theory and realistic superconducting-qubit layouts (IBM’s 5-qubit and 127-qubit processors) is particularly valuable.

Weaknesses

While the main theorem is solid and well-presented, several aspects could be better contextualized or expanded.

1) The novelty of the result compared with earlier compositional controllability studies (e.g., Zeier & Schulte-Herbrüggen 2011; Albertini & D’Alessandro 2025) is not sufficiently emphasized. 2) The discussion about generalization to qudits is too brief and lacks insight into the main technical obstacles or conceptual challenges. 3)The final section could be more clearly structured and expanded to better integrate perspectives on qudit generalization and operational costs.

Report

The article is a strong contribution to the field of quantum control, with immediate relevance for modular and scalable hardware architectures. The theoretical formalism is carefully constructed and accessible. The examples grounded in superconducting architectures add credibility and concreteness.

Nevertheless, a few areas would benefit from additional clarification:

1) A more explicit comparison with previous compositional controllability results, highlighting the precise novelty of the present theorem. 2) An expanded discussion or appendix outlining either the generalization path or the open questions for qudit systems. 3) A more quantitative treatment of scalability — for instance, how the depth of nested commutators or computational effort scales with the number of subsystems. 4) A reorganization of Section IV into “Summary and Perspectives,” including the potential to discuss quantum speed limits and operational trade-offs (gate time, fidelity, decoherence) when connectivity is reduced.

Requested changes

1) Add a discussion of the main novelty compared to prior compositional controllability work (Zeier & Schulte-Herbrüggen 2011; Albertini & D’Alessandro 2025). 2) Deepen the discussion on the extension to qudits — at least qualitatively describing the obstacles or modifications needed. 3) Provide a brief quantitative or conceptual analysis of scaling properties (e.g., commutator depth, computational complexity). 4) Rename Section IV to “Summary and Perspectives” (or simply “Discussion”) and integrate the extended discussion on qudits and operational costs. 5)Correct the typo: “must me linearly independent” → “must be linearly independent.”

Recommendation

Publish (easily meets expectations and criteria for this Journal; among top 50%)

---

## Round 2 · Referee Report · Anonymous (Referee 1) · 2025-11-4

Report

This article adresses the question of global controllability of large qubit arrays. The main result is a template to design modular Quantum Processing Units composed of a collection of smaller units that are known to be controllable. The controllability is achieved by inserting an entangling control coupling two qubits located in two neighbouring subunits. The proposed architecture is illustrated with the examples of arrays of 10 or 127 qubits inspired by IBM's quantum processors. The controllability of the full array is proven using well-established methods based on Lie algebras, that the authors have used in earlier articles.

The article is structures in two parts: The first one is a qualitative description of the problem and of the main elements of the proposed architecture. The second par, written as an appendix contains the precise technical formulation and the proofs of the announced results. The authors discuss carefully the role of the controllability proofs, remarking that other properties, like the attainable speed, will also have to be taken into account in order to establish the viability and efficiency of a given design.

The article is clearly written and the results are novel and interesting. I recommend the publication of this article in the journal SciPost.

The authors may consider the following suggestions:

1) The notation with X, Y, Z in Eqs (3) and (4) should be defined or changed into $\sigma$'s.

2) It would be useful to add a comment on how the entangling control, e.g. a tunable coupler, can be implemented on a machine like IBM's computer.

3) A comment could be added concerning the kind of quantum algorithms could need or profit from unitary operations acting on 127 qubits.

Recommendation

Publish (surpasses expectations and criteria for this Journal; among top 10%)

---

## Round 2 · Referee Report · Anonymous (Referee 3) · 2025-11-11

Report

The authors prove using Lie algebraic arguments that, if two fully controllable quantum systems composed of qubits are coupled with a tunable interaction entangling any qubit from the first with any qubit of the second, then the composite system is also fully controllable.They use this theoretical result to demonstrate how the controls and couplings in some IBM quantum processors can be considerably reduced, while the corresponding systems remain fully controllable. The authors recognize that reducing the number of controls might slow down the speed of the computations. The article is very clear and well written, the examples present potential useful applications of the theoretical result, and the proof of the main theorem is straightforward. The work is timely and interesting, opening the discussion about how much control can be lost in a quantum system without reducing much the speed of the calculations. For these reasons we believe that it warrants publication in SciPost Physics. We only have some minor comments, mostly typos spotted in the text and appendix:

A. For the composite system with drift the sum of the individual drifts, the authors prove that the individual drifts belong to the Lie algebra. If the coupling of the individual systems can take the zero value, isn't this observation straightforward?

B. Typos in main text: 

  1. Page 3, end of first paragraph "its use it for", please rephrase.
  2. In the text below Eq. (5d) is H_{5T}, H_{5L}, H_{4L}.
  3. The word "module" appears several times in the text withιν quotation marks; the first quotation mark does not have the right direction. 

C. Typos in Appendix: 1. End of page c, "extracted from described in Chapter 3", please rephrase. 2. Lemma3, second line, place a comma between the operators in the commutator. 3. Page e, before Eqs. (A15), (A16) the authors discuss a product of \sigma_3 matrices but in (A16) end up with a product of \sigma_1 matrices. Is it ok or a typo? 4. In the first line of (A20) the first operator should not be 1 but H_0^{AB}. 5. Page f, two lines below the white square, the operator reduced to a single product is H_c^{\mu, n}, not H_0^{AB}. Same page, before Eq. (A20), "Equation Eq. (A5)", delete "Equation".

Recommendation

Publish (meets expectations and criteria for this Journal)

---

## Round 3 · Author Response

Dear editors, dear reviewers,

we are writing in response to the editorial decision made on our article “Scalable modular architecture for universal quantum computation”.

We would like to thank the reviewers for their time and effort spent on our manuscript, the positive assessments and helpful comments they have provided. A new version of the manuscript may be found on arXiv ( arXiv:2507.14691v3 ).

We believe that with these revisions our manuscript may be accepted for publication in SciPost Physics.

Sincerely yours,
Fernando Gago-Encinas, and Christiane P. Koch

---

## Round 3 · List of Changes

Changes following report #1

The authors thank the reviewer for their appraisal and feedback. Here is a complete list of the corrections in response to the reviewer’s suggestions.

1) The change of notation has been implemented to match the rest of the article.

2) A comment has been added in Section II to reference some of the latest works on tunable couplings in superconducting qubits and in particular in IBM’s QPUs, complementing the existing references to more conceptual works. 3) A sentence has been added to the beginning of Section III, mentioning algorithms that benefit from a large number of qubits.

Changes following report #2

The authors thank the reviewer for their valuable insights. The typo has been corrected. Here we provide a detailed list of changes to the manuscript following the reviewer’s suggestions:

1) The article has been better put into context by explicitly stating the differences between the result here shown and previous scientific contributions, in particular Zeier & Schulte-Herbrüggen 2011 and Albertini & D’Alessandro 2025. This clarification is found in Section II, after Theorem 1.

2) The discussion of the extensionof the current proof to the case of qudits has been expanded in the last section of the article. In particular, itis pointed out that the result also holds thanks to previous theoretical results and the only missing part is the exact operations needed.

3) We agree that a quantitative analysis of how computational efforts scale with the number of subsystems would be a very interesting topic. However, this question merits a study of its own, far exceeding the scope of the work here presented. Whether studying the problem via quantum speed limits or via the quantum circuit depth after transpilation, the required optimizations need careful tuning and quickly become complex for larger systems.

4) Section IV has been renamed and expanded. In particular, we have added a discussion of our insights regarding quantum speed limit, circuit depth and fidelity.

Changes following report #3

A) Surprisingly, it is not completely straightforward since it is always the joint drift that governs the evolution, even with zero coupling, which implies that the individual drifts H_0,A and H_0,B evolve for the same amount of time. In the particular case where both drifts have the same period, it becomes necessary to prove that they can be decoupled.

C.3) The indices are indeed correct. We start with the term with \sigma_3 but due to the commutator rules it ends up as \sigma_1. However, note that the coefficient in front of it is still c_{3,3}, showing that the original term was the product of two \sigma_3 matrices.

---

## Editorial Decision

in_refereeing